# Variational Denoising Network: Toward Blind Noise Modeling and Removal

Zongsheng Yue[1,2], Hongwei Yong[2], Qian Zhao[1], Lei Zhang[2,3], Deyu Meng[4,1,*]

[1] *School of Mathematics and Statistics, Xi'an Jiaotong University, Shaanxi, China*
[2]*Department of Computing, Hong Kong Polytechnic University, Kowloon, Hong Kong*
[3]*DAMO Academy, Alibaba Group, Shenzhen, China*
[4]*Faculty of Information Technology, The Macau University of Science and Technology, Macau, China*
[*]*Corresponding author: dymeng@mail.xjtu.edu.cn*

## Abstract

Blind image denoising is an important yet very challenging problem in computer vision due to the complicated acquisition process of real images. In this work we propose a new variational inference method, which integrates both noise estimation and image denoising into a unique Bayesian framework, for blind image denoising. Specifically, an approximate posterior, parameterized by deep neural networks, is presented by taking the intrinsic clean image and noise variances as latent variables conditioned on the input noisy image. This posterior provides explicit parametric forms for all its involved hyper-parameters, and thus can be easily implemented for blind image denoising with automatic noise estimation for the test noisy image. On one hand, as other data-driven deep learning methods, our method, namely variational denoising network (VDN), can perform denoising efficiently due to its explicit form of posterior expression. On the other hand, VDN inherits the advantages of traditional model-driven approaches, especially the good generalization capability of generative models. VDN has good interpretability and can be flexibly utilized to estimate and remove complicated non-i.i.d. noise collected in real scenarios. Comprehensive experiments are performed to substantiate the superiority of our method in blind image denoising.

## 1 Introduction

Image denoising is an important research topic in computer vision, aiming at recovering the underlying clean image from an observed noisy one. The noise contained in a real noisy image is generally accumulated from multiple different sources, e.g., capturing instruments, data transmission media, image quantization, etc. [40]. Such complicated generation process makes it fairly difficult to access the noise information accurately and recover the underlying clean image from the noisy one. This constitutes the main aim of blind image denoising.

There are two main categories of image denoising methods. Most classical methods belong to the first category, mainly focusing on constructing a rational maximum a posteriori (MAP) model, involving the fidelity (loss) and regularization terms, from a Bayesian perspective [6]. An understanding for data generation mechanism is required for designing a rational MAP objective, especially better image priors like sparsity [3], low-rankness [16, 50, 42], and non-local similarity [9, 27]. These methods are superior mainly in their interpretability naturally led by the Bayesian framework. They, however, still exist critical limitations due to their assumptions on both image prior and noise (generally i.i.d. Gaussian), possibly deviating from real spatially variant (i.e.,non-i.i.d.) noise, and their relatively low implementation speed since the algorithm needs to be re-implemented for any new coming image. Recently, deep learning approaches represent a new trend along this research line. The main idea is to firstly collect large amount of noisy-clean image pairs and then train a deep neural network denoiser on these training data in an end-to-end learning manner. This approach is especially superior in its

effective accumulation of knowledge from large datasets and fast denoising speed for test images. They, however, are easy to overfit to the training data with certain noisy types, and still could not be generalized well on test images with unknown but complicated noises.

Thus, blind image denoising especially for real images is still a challenging task, since the real noise distribution is difficult to be pre-known (for model-driven MAP approaches) and hard to be comprehensively simulated by training data (for data-driven deep learning approaches).

Against this issue, this paper proposes a new variational inference method, aiming at directly inferring both the underlying clean image and the noise distribution from an observed noisy image in a unique Bayesian framework. Specifically, an approximate posterior is presented by taking the intrinsic clean image and noise variances as latent variables conditioned on the input noisy image. This posterior provides explicit parametric forms for all its involved hyper-parameters, and thus can be efficiently implemented for blind image denoising with automatic noise estimation for test noisy images.

In summary, this paper mainly makes following contributions: 1) The proposed method is capable of simultaneously implementing both noise estimation and blind image denoising tasks in a unique Bayesian framework. The noise distribution is modeled as a general non-i.i.d. configurations with spatial relevance across the image, which evidently better complies with the heterogeneous real noise beyond the conventional i.i.d. noise assumption. 2) Succeeded from the fine generalization capability of the generative model, the proposed method is verified to be able to effectively estimate and remove complicated non-i.i.d. noises in test images even though such noise types have never appeared in training data, as clearly shown in Fig. 3. 3) The proposed method is a generative approach outputted a complete distribution revealing how the noisy image is generated. This not only makes the result with more comprehensive interpretability beyond traditional methods purely aiming at obtaining a clean image, but also naturally leads to a learnable likelihood (fidelity) term according to the data-self. 4) The most commonly utilized deep learning paradigm, i.e., taking MSE as loss function and training on large noisy-clean image pairs, can be understood as a degenerated form of the proposed generative approach. Their overfitting issue can then be easily explained under this variational inference perspective: these methods intrinsically put dominant emphasis on fitting the priors of the latent clean image, while almost neglects the effect of noise variations. This makes them incline to overfit noise bias on training data and sensitive to the distinct noises in test noisy images.

The paper is organized as follows: Section 2 introduces related work. Sections 3 presents the proposed full Bayesion model, the deep variational inference algorithm, the network architecture and some discussions. Section 4 demonstrates experimental results and the paper is finally concluded.

## 2 Related Work

We present a brief review for the two main categories of image denoising methods, i.e., model-driven MAP based methods and data-driven deep learning based methods.

**Model-driven MAP based Methods:** Most classical image denoising methods belong to this category, through designing a MAP model with a fidelity/loss term and a regularization one delivering the pre-known image prior. Along this line, total variation denoising [37], anisotropic diffusion [31] and wavelet coring [38] use the statistical regularities of images to remove the image noise. Later, the nonlocal similarity prior, meaning many small patches in a non-local image area possess similar configurations, was widely used in image denoising. Typical ones include CBM3D [11] and non-local means [9]. Some dictionary learning methods [16, 13, 42] and Field-of-Experts (FoE) [36], also revealing certain prior knowledge of image patches, had also been attempted for the task. Several other approaches focusing on the fidelity term, which are mainly determined by the noise assumption on data. E.g., Mulitscale [23] assumed the noise of each patch and its similar patches in the same image to be correlated Gaussian distribution, and LR-MoG [30, 48, 50], DP-GMM [44] and DDPT [49] fitted the image noise by using Mixture of Gaussian (MoG) as an approximator for noises.

**Data-driven Deep Learning based Methods:** Instead of pre-setting image prior, deep learning methods directly learn a denoiser (formed as a deep neural network) from noisy to clean ones on a large collection of noisy-clean image pairs. Jain and Seung [19] firstly adopted a five layer convolution neural network (CNN) for the task. Then some auto-encoder based methods [41, 2] were applied. Meantime, Burger et al. [10] achieved the comparable performance with BM3D using plain multi-layer perceptron (MLP). Zhang et al. [45] further proposed the denoising convolution network (DnCNN) and achieved state-of-the-art performance on Gaussian denoising tasks. Mao et al. [29] proposed a deep fully convolution encoding-decoding network with symmetric skip connection. Tai

et al. [39] preposed a very deep persistent memory network (MemNet) to explicitly mine persistent memory through an adaptive learning process. Recently, NLRN [25], N3Net [33] and UDNet [24] all embedded the non-local property of image into DNN to facilitate the denoising task. In order to boost the flexibility against spatial variant noise, FFDNet [46] was proposed by pre-evaluating the noise level and inputting it to the network together with the noisy image. Guo et al. [17] and Brooks et al. [8] both attempted to simulate the generation process of the images in camera.

## 3  Variational Denoising Network for Blind Noise Modeling

Given training set $D = \{\boldsymbol{y}_j, \boldsymbol{x}_j\}_{j=1}^n$, where $\boldsymbol{y}_j, \boldsymbol{x}_j$ denote the $j^{th}$ training pair of noisy and the expected clean images, $n$ represents the number of training images, our aim is to construct a variational parametric approximation to the posterior of the latent variables, including the latent clean image and the noise variances, conditioned on the noisy image. Note that for the noisy image $\boldsymbol{y}$, its training pair $\boldsymbol{x}$ is generally a simulated "clean" one obtained as the average of many noisy ones taken under similar camera conditions [4, 1], and thus is always not the exact latent clean image $\boldsymbol{z}$. This explicit parametric posterior can then be used to directly infer the clean image and noise distribution from any test noisy image. To this aim, we first need to formulate a rational full Bayesian model of the problem based on the knowledge delivered by the training image pairs.

### 3.1  Constructing Full Bayesian Model Based on Training Data

Denote $\boldsymbol{y} = [y_1, \cdots, y_d]^T$ and $\boldsymbol{x} = [x_1, \cdots, x_d]^T$ as any training pair in $D$, where $d$ (width*height) is the size of a training image[1]. We can then construct the following model to express the generation process of the noisy image $\boldsymbol{y}$:

$$y_i \sim \mathcal{N}(y_i | z_i, \sigma_i^2), \ i = 1, 2, \cdots, d, \tag{1}$$

where $\boldsymbol{z} \in \mathbb{R}^d$ is the latent clean image underlying $\boldsymbol{y}$, $\mathcal{N}(\cdot | \mu, \sigma^2)$ denotes the Gaussian distribution with mean $\mu$ and variance $\sigma^2$. Instead of assuming i.i.d. distribution for the noise as conventional [28, 13, 16, 42], which largely deviates the spatial variant and signal-depend characteristics of the real noise [46, 8], we models the noise as a non-i.i.d. and pixel-wise Gaussian distribution in Eq. (1).

The simulated "clean" image $\boldsymbol{x}$ evidently provides a strong prior to the latent variable $\boldsymbol{z}$. Accordingly we impose the following conjugate Gaussian prior on $\boldsymbol{z}$:

$$z_i \sim \mathcal{N}(z_i | x_i, \varepsilon_0^2), \ i = 1, 2, \cdots, d, \tag{2}$$

where $\varepsilon_0$ is a hyper-parameter and can be easily set as a small value.

Besides, for $\boldsymbol{\sigma}^2 = \{\sigma_1^2, \sigma_2^2, \cdots, \sigma_d^2\}$, we also introduce a rational conjugate prior as follows:

$$\sigma_i^2 \sim \text{IG}\left(\sigma_i^2 | \frac{p^2}{2} - 1, \frac{p^2 \xi_i}{2}\right), \ i = 1, 2, \cdots, d, \tag{3}$$

where $\text{IG}(\cdot | \alpha, \beta)$ is the inverse Gamma distribution with parameter $\alpha$ and $\beta$, $\boldsymbol{\xi} = \mathcal{G}\left((\hat{\boldsymbol{y}} - \hat{\boldsymbol{x}})^2; p\right)$ represents the filtering output of the variance map $(\hat{\boldsymbol{y}} - \hat{\boldsymbol{x}})^2$ by a Gaussian filter with $p \times p$ window, and $\hat{\boldsymbol{y}}, \hat{\boldsymbol{x}} \in \mathbb{R}^{h \times w}$ are the matrix (image) forms of $\boldsymbol{y}, \boldsymbol{x} \in \mathbb{R}^d$, respectively. Note that the mode of above IG distribution is $\xi_i$ [6, 43], which is a approximate evaluation of $\sigma_i^2$ in $p \times p$ window.

Combining Eqs. (1)-(3), a full Bayesian model for the problem can be obtained. The goal then turns to infer the posterior of latent variables $\boldsymbol{z}$ and $\boldsymbol{\sigma}^2$ from noisy image $\boldsymbol{y}$, i.e., $p(\boldsymbol{z}, \boldsymbol{\sigma}^2 | \boldsymbol{y})$.

### 3.2  Variational Form of Posterior

We first construct a variational distribution $q(\boldsymbol{z}, \boldsymbol{\sigma}^2 | \boldsymbol{y})$ to approximate the posterior $p(\boldsymbol{z}, \boldsymbol{\sigma}^2 | \boldsymbol{y})$ led by Eqs. (1)-(3). Similar to the commonly used mean-field variation inference techniques, we assume conditional independence between variables $\boldsymbol{z}$ and $\boldsymbol{\sigma}^2$, i.e.,

$$q(\boldsymbol{z}, \boldsymbol{\sigma}^2 | \boldsymbol{y}) = q(\boldsymbol{z} | \boldsymbol{y}) q(\boldsymbol{\sigma}^2 | \boldsymbol{y}). \tag{4}$$

Based on the conjugate priors in Eqs. (2) and (3), it is natural to formulate variational posterior forms of $\boldsymbol{z}$ and $\boldsymbol{\sigma}^2$ as follows:

$$q(\boldsymbol{z} | \boldsymbol{y}) = \prod_i^d \mathcal{N}(z_i | \mu_i(\boldsymbol{y}; W_D), m_i^2(\boldsymbol{y}; W_D)), \ q(\boldsymbol{\sigma}^2 | \boldsymbol{y}) = \prod_i^d \text{IG}(\sigma_i^2 | \alpha_i(\boldsymbol{y}; W_S), \beta_i(\boldsymbol{y}; W_S)), \tag{5}$$

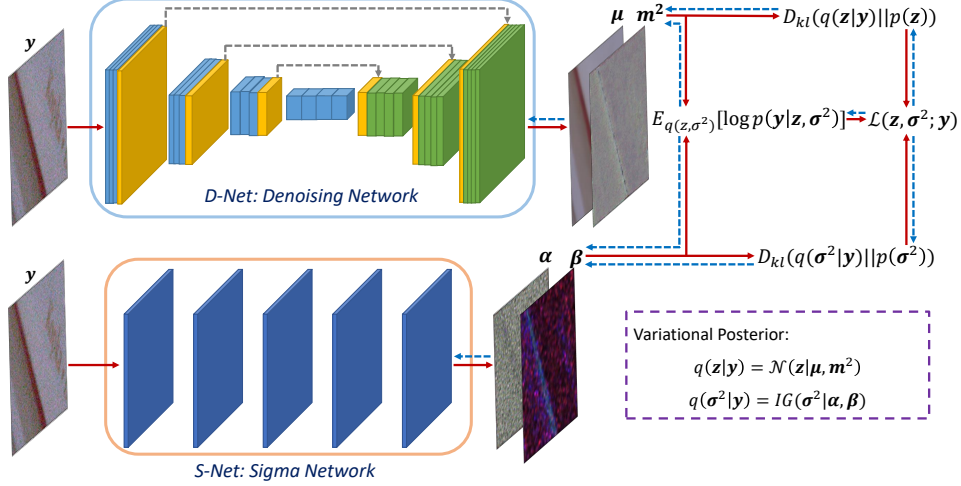

Figure 1: The architecture of the proposed deep variational inference network. The red solid lines denote the forward process, and the blue dotted lines mark the gradient flow direction in the BP algorithm.

where $\mu_i(\boldsymbol{y}; W_D)$ and $m_i^2(\boldsymbol{y}; W_D)$ are designed as the prediction functions for getting posterior parameters of latent variable $\boldsymbol{z}$ directly from $\boldsymbol{y}$. The function is represented as a network, called denoising network or *D-Net*, with parameters $W_D$. Similarly, $\alpha_i(\boldsymbol{y}; W_S)$ and $\beta_i(\boldsymbol{y}; W_S)$ denote the prediction functions for evaluating posterior parameters of $\boldsymbol{\sigma}^2$ from $\boldsymbol{y}$, where $W_S$ represents the parameters of the network, called Sigma network or *S-Net*. The aforementioned is illustrated in Fig. 1. Our aim is then to optimize these network parameters $W_D$ and $W_S$ so as to get the explicit functions for predicting clean image $\boldsymbol{z}$ as well as noise knowledge $\boldsymbol{\sigma}^2$ from any test noisy image $\boldsymbol{y}$. A rational objective function with respect to $W_D$ and $W_S$ is thus necessary to train both the networks.

Note that the network parameters $W_D$ and $W_S$ are shared by posteriors calculated on all training data, and thus if we train them on the entire training set, the method is expected to induce the general statistical inference insight from noisy image to its underlying clean image and noise level.

### 3.3 Variational Lower Bound of Marginal Data Likelihood

For notation convenience, we simply write $\mu_i(\boldsymbol{y}; W_D)$, $m_i^2(\boldsymbol{y}; W_D)$, $\alpha_i(\boldsymbol{y}; W_S)$, $\beta_i(\boldsymbol{y}; W_S)$ as $\mu_i$, $m_i^2$, $\alpha_i$, $\beta_i$ in the following calculations. For any noisy image $\boldsymbol{y}$ and its simulated "clean" image $\boldsymbol{x}$ in the training set, we can decompose its marginal likelihood as the following form [7]:

$$\log p(\boldsymbol{y}; \boldsymbol{z}, \boldsymbol{\sigma}^2) = \mathcal{L}(\boldsymbol{z}, \boldsymbol{\sigma}^2; \boldsymbol{y}) + D_{KL}\left(q(\boldsymbol{z}, \boldsymbol{\sigma}^2|\boldsymbol{y})||p(\boldsymbol{z}, \boldsymbol{\sigma}^2|\boldsymbol{y})\right), \tag{6}$$

where

$$\mathcal{L}(\boldsymbol{z}, \boldsymbol{\sigma}^2; \boldsymbol{y}) = E_{q(\boldsymbol{z}, \boldsymbol{\sigma}^2|\boldsymbol{y})}\left[\log p(\boldsymbol{y}|\boldsymbol{z}, \boldsymbol{\sigma}^2)p(\boldsymbol{z})p(\boldsymbol{\sigma}^2) - \log q(\boldsymbol{z}, \boldsymbol{\sigma}^2|\boldsymbol{y})\right], \tag{7}$$

Here $E_{p(x)}[f(x)]$ represents the exception of $f(x)$ w.r.t. stochastic variable $x$ with probability density function $p(x)$. The second term of Eq. (6) is a KL divergence between the variational approximate posterior $q(\boldsymbol{z}, \boldsymbol{\sigma}^2|\boldsymbol{y})$ and the true posterior $p(\boldsymbol{z}, \boldsymbol{\sigma}^2|\boldsymbol{y})$ with non-negative value. Thus the first term $\mathcal{L}(\boldsymbol{z}, \boldsymbol{\sigma}^2; \boldsymbol{y})$ constitutes a *variational lower bound* on the marginal likelihood of $p(\boldsymbol{y}|\boldsymbol{z}, \boldsymbol{\sigma}^2)$, i.e.,

$$\log p(\boldsymbol{y}; \boldsymbol{z}, \boldsymbol{\sigma}^2) \geq \mathcal{L}(\boldsymbol{z}, \boldsymbol{\sigma}^2; \boldsymbol{y}). \tag{8}$$

According to Eqs. (4), (5) and (7), the lower bound can then be rewritten as:

$$\mathcal{L}(\boldsymbol{z}, \boldsymbol{\sigma}^2; \boldsymbol{y}) = E_{q(\boldsymbol{z}, \boldsymbol{\sigma}^2|\boldsymbol{y})}\left[\log p(\boldsymbol{y}|\boldsymbol{z}, \boldsymbol{\sigma}^2)\right] - D_{KL}\left(q(\boldsymbol{z}|\boldsymbol{y})||p(\boldsymbol{z})\right) - D_{KL}\left(q(\boldsymbol{\sigma}^2|\boldsymbol{y})||p(\boldsymbol{\sigma}^2)\right). \tag{9}$$

It's pleased that all the three terms in Eq (9) can be integrated analytically as follows:

$$E_{q(\boldsymbol{z}, \boldsymbol{\sigma}^2|\boldsymbol{y})}\left[\log p(\boldsymbol{y}|\boldsymbol{z}, \boldsymbol{\sigma}^2)\right] = \sum_{i=1}^{d}\left\{-\frac{1}{2}\log 2\pi - \frac{1}{2}(\log \beta_i - \psi(\alpha_i)) - \frac{\alpha_i}{2\beta_i}\left[(y_i - \mu_i)^2 + m_i^2\right]\right\}, \tag{10}$$

$$D_{KL}\left(q(\boldsymbol{z}|\boldsymbol{y})||p(\boldsymbol{z})\right) = \sum_{i=1}^{d}\left\{\frac{(\mu_i - x_i)^2}{2\varepsilon_0^2} + \frac{1}{2}\left[\frac{m_i^2}{\varepsilon_0^2} - \log\frac{m_i^2}{\varepsilon_0^2} - 1\right]\right\}, \tag{11}$$

$$D_{KL}\left(q(\boldsymbol{\sigma}^2|\boldsymbol{y})||p(\boldsymbol{\sigma^2})\right) = \sum_{i=1}^{d}\left\{\left(\alpha_i - \frac{p^2}{2} + 1\right)\psi(\alpha_i) + \left[\log\Gamma\left(\frac{p^2}{2}-1\right) - \log\Gamma(\alpha_i)\right]\right.$$
$$\left. + \left(\frac{p^2}{2}-1\right)\left(\log\beta_i - \log\frac{p^2\xi_i}{2}\right) + \alpha_i\left(\frac{p^2\xi_i}{2\beta_i}-1\right)\right\}, \quad (12)$$

where $\psi(\cdot)$ denotes the digamma function. Calculation details are listed in supplementary material.

We can then easily get the expected objective function (i.e., a negtive lower bound of the marginal likelihood on entire training set) for optimizing the network parameters of *D-Net* and *S-Net* as follows:

$$\min_{W_D, W_S} - \sum_{j=1}^{n}\mathcal{L}(\boldsymbol{z}_j, \boldsymbol{\sigma}_j^2; \boldsymbol{y}_j). \quad (13)$$

### 3.4 Network Learning

As aforementioned, we use *D-Net* and *S-Net* together to infer the variational parameters $\boldsymbol{\mu}$, $\boldsymbol{m}^2$ and $\boldsymbol{\alpha}$, $\boldsymbol{\beta}$ from the input noisy image $\boldsymbol{y}$, respectively, as shown in Fig. 1. It is critical to consider how to calculate derivatives of this objective with respect to $W_D, W_S$ involved in $\boldsymbol{\mu}$, $\boldsymbol{m}^2$, $\boldsymbol{\alpha}$ and $\boldsymbol{\beta}$ to facilitate an easy use of stochastic gradient varitional inference. Fortunately, different from other related variational inference techniques like VAE [22], all three terms of Eqs. (10)-(12) in the lower bound Eq. (9) are differentiable and their derivatives can be calculated analytically without the need of any reparameterization trick, largely reducing the difficulty of network training.

At the training stage of our method, the network parameters can be easily updated with backprop-agation (BP) algorithm [15] through Eq. (13). The function of each term in this objective can be intuitively explained: the first term represents the likelihood of the observed noisy images in training set, and the last two terms control the discrepancy between the variational posterior and the corre-sponding prior. During the BP training process, the gradient information from the likelihood term of Eq. (10) is used for updating both the parameters of *D-Net* and *S-Net* simultaneously, implying that the inference for the latent clean image $\boldsymbol{z}$ and $\boldsymbol{\sigma}^2$ is guided to be learned from each other.

At the test stage, for any test noisy image, through feeding it into *D-Net*, the final denoising result can be directly obtained by $\boldsymbol{\mu}$. Additionally, through inputting the noisy image to the *S-Net*, the noise distribution knowledge (i.e., $\boldsymbol{\sigma}^2$) is easily inferred. Specifically, the noise variance in each pixel can be directly obtained by using the mode of the inferred inverse Gamma distribution: $\sigma_i^2 = \frac{\beta_i}{(\alpha_i+1)}$.

### 3.5 Network Architecture

The D-Net in Fig. 1 takes the noisy image $\boldsymbol{y}$ as input to infer the variational parameters $\boldsymbol{\mu}$ and $\boldsymbol{m}^2$ in $q(\boldsymbol{z}|\boldsymbol{y})$ of Eq. (5), and performs the denoising task in the proposed variational inference algorithm. In order to capture multi-scale information of the image, we use a U-Net [35] with depth 4 as the D-Net, which contains 4 encoder blocks ([*Conv+ReLU*]×2+*Average pooling*), 3 decoder blocks (*Transpose Conv*+[*Conv+ReLU*]×2) and symmetric skip connection under each scale. For parameter $\boldsymbol{\mu}$, the residual learning strategy is adopted as in [45], i.e., $\boldsymbol{\mu} = \boldsymbol{y} + f(\boldsymbol{y}; W_D)$, where $f(\cdot; W_D)$ denotes the *D-Net* with parameters $W_D$. As for the *S-Net*, which takes the noisy image $\boldsymbol{y}$ as input and outputs the predicted variational parameters $\boldsymbol{\alpha}$ and $\boldsymbol{\beta}$ in $q(\boldsymbol{\sigma}^2|\boldsymbol{y})$ of Eq (5), we use the DnCNN [45] architecture with five layers, and the feature channels of each layer is set as 64. It should be noted that our proposed method is a general framework, most of the commonly used network architectures [46, 34, 24, 47] in image restoration can also be easily substituted.

### 3.6 Some Discussions

It can be seen that the proposed method succeeds advantages of both model-driven MAP and data-driven deep learning methods. On one hand, our method is a generative approach and possesses fine interpretability to the data generation mechanism; and on the other hand it conducts an explicit prediction function, facilitating efficient image denoising as well as noise estimation directly through an input noisy image. Furthermore, beyond current methods, our method can finely evaluate and remove non-i.i.d. noises embedded in images, and has a good generalization capability to images with complicated noises, as evaluated in our experiments. This complies with the main requirement of the blind image denoising task.

If we set the hyper-parameter $\varepsilon_0^2$ in Eq.(2) as an extremely small value close to 0, it is easy to see that the objective of the proposed method is dominated by the second term of Eq. (10), which makes

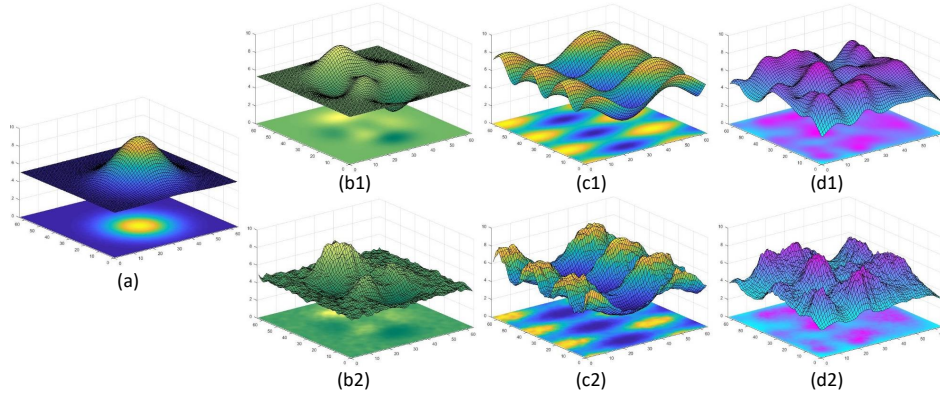

Figure 2: (a) The spatially variant map $M$ for noise generation in training data. (b1)-(d1): Three different $M$s on testing data in Cases 1-3. (b2)-(d2): Correspondingly predicted $M$s by our method on the testing data.

Table 1: The PSNR(dB) results of all competing methods on the three groups of test datasets. The best and second best results are highlighted in bold and Italic, respectively.

| Cases | Datasets | Methods | | | | | | | | | |
|-------|----------|-------|------|------|-----|--------|--------|--------|-----------|-------|-----|
| | | CBM3D | WNNM | NCSR | MLP | DnCNN-B | MemNet | FFDNet | FFDNet$_v$ | UDNet | VDN |
| Case 1 | Set5 | 27.76 | 26.53 | 26.62 | 27.26 | 29.85 | 30.10 | *30.16* | 30.15 | 28.13 | **30.39** |
| | LIVE1 | 26.58 | 25.27 | 24.96 | 25.71 | 28.81 | 28.96 | *28.99* | 28.96 | 27.19 | **29.22** |
| | BSD68 | 26.51 | 25.13 | 24.96 | 25.58 | 28.73 | 28.74 | *28.78* | 28.77 | 27.13 | **29.02** |
| Case 2 | Set5 | 26.34 | 24.61 | 25.76 | 25.73 | 29.04 | 29.55 | *29.60* | 29.56 | 26.01 | **29.80** |
| | LIVE1 | 25.18 | 23.52 | 24.08 | 24.31 | 28.18 | 28.56 | *28.58* | 28.56 | 25.25 | **28.82** |
| | BSD68 | 25.28 | 23.52 | 24.27 | 24.30 | 28.15 | 28.36 | *28.43* | 28.42 | 25.13 | **28.67** |
| Case 3 | Set5 | 27.88 | 26.07 | 26.84 | 26.88 | 29.13 | 29.51 | *29.54* | 29.49 | 27.54 | **29.74** |
| | LIVE1 | 26.50 | 24.67 | 24.96 | 25.26 | 28.17 | 28.37 | *28.39* | 28.38 | 26.48 | **28.65** |
| | BSD68 | 26.44 | 24.60 | 24.95 | 25.10 | 28.11 | 28.20 | *28.22* | 28.20 | 26.44 | **28.46** |

the objective degenerate as the MSE loss generally used in traditional deep learning methods (i.e., minimizing $\sum_{j=1}^{n} ||\boldsymbol{\mu}(\boldsymbol{y}_j; W_D) - \boldsymbol{x}_j||^2$. This provides a new understanding to explain why they incline to overfit noise bias in training data. The posterior inference process puts dominant emphasis on fitting priors imposed on the latent clean image, while almost neglects the effect of noise variations. This naturally leads to its sensitiveness to unseen complicated noises contained in test images.

Very recently, both CBDNet [17] and FFDNet [46] are presented for the denoising task by feeding the noisy image integrated with the pre-estimated noise level into the deep network to make it better generalize to distinct noise types in training stage. Albeit more or less improving the generalization capability of network, such strategy is still too heuristic and is not easy to interpret how the input noise level intrinsically influence the final denoising result. Comparatively, our method is constructed in a sound Bayesian manner to estimate clean image and noise distribution together from the input noisy image, and its generalization can be easily explained from the perspective of generative model.

## 4 Experimental Results

We evaluate the performance of our method on synthetic and real datasets in this section. All experiments are evaluated in the sRGB space. We briefly denote our method as VDN in the following. The training and testing codes of our VDN is available at `https://github.com/zsyOAOA/VDNet`.

### 4.1 Experimental Setting

**Network training and parameter setting:** The weights of *D-Net* and *S-Net* in our variational algorithm were initialized according to [18]. In each epoch, we randomly crop $N = 64 \times 5000$ patches with size $128 \times 128$ from the images for training. The Adam algorithm [21] is adopted to optimize the network parameters through minimizing the proposed negative lower bound objective. The initial learning rate is set as $2e\text{-}4$ and linearly decayed in half every 10 epochs until to $1e\text{-}6$. The window size $p$ in Eq. (3) is set as 7. The hyper-parameter $\varepsilon_0^2$ is set as $5e\text{-}5$ and $1e\text{-}6$ in the following synthetic and real-world image denoising experiments, respectively.

**Comparison methods:** Several state-of-the-art denoising methods are adopted for performance comparison, including CBM3D [11], WNNM [16], NCSR [14], MLP [10], DnCNN-B [45], MemNet [39], FFDNet [46], UDNet [24] and CBDNet [17]. Note that CBDNet is mainly designed for blind denoising task, and thus we only compared CBDNet on the real noise removal experiments.

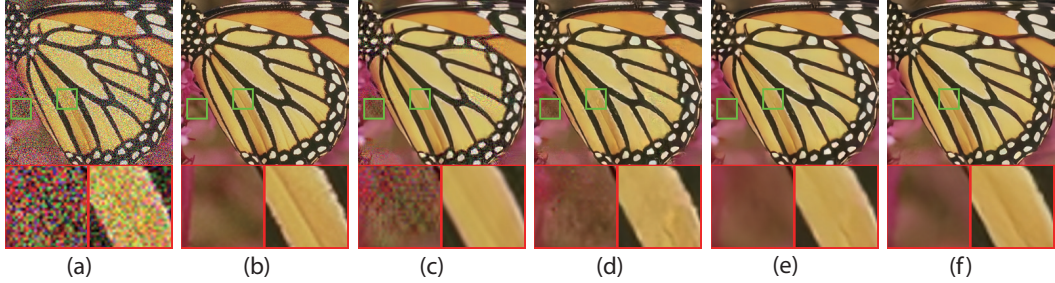

$$\begin{array}{cccccc} \text{(a)} & \text{(b)} & \text{(c)} & \text{(d)} & \text{(e)} & \text{(f)} \end{array}$$

Figure 3: Image denoising results of a typical test image in Case 2. (a) Noisy image, (b) Groundtruth, (c) CBM3D (24.63dB), (d) DnCNN-B (27.83dB), (e) FFDNet (28.06dB), (f) VDN (28.32dB).

Table 2: The PSNR(dB) results of all competing methods on AWGN noise cases of three test datasets.

| Sigma | Datasets | Methods | | | | | | | | | |
|---|---|---|---|---|---|---|---|---|---|---|---|
| | | CBM3D | WNNM | NCSR | MLP | DnCNN-B | MemNet | FFDNet | FFDNet$_e$ | UDNet | VDN |
| $\sigma = 15$ | Set5 | 33.42 | 32.92 | 32.57 | - | 34.04 | 34.18 | 34.30 | *34.31* | 34.19 | **34.34** |
| | LIVE1 | 32.85 | 31.70 | 31.46 | - | 33.72 | 33.84 | **33.96** | **33.96** | 33.74 | 33.94 |
| | BSD68 | 32.67 | 31.27 | 30.84 | - | *33.87* | 33.76 | 33.85 | 33.68 | 33.76 | **33.90** |
| $\sigma = 25$ | Set5 | 30.92 | 30.61 | 30.33 | 30.55 | 31.88 | 31.98 | *32.10* | 32.09 | 31.82 | **32.24** |
| | LIVE1 | 30.05 | 29.15 | 29.05 | 29.16 | 31.23 | 31.26 | *31.37* | *31.37* | 31.09 | **31.50** |
| | BSD68 | 29.83 | 28.62 | 28.35 | 28.93 | 31.17 | 31.17 | *31.22* | 31.21 | 31.02 | **31.35** |
| $\sigma = 50$ | Set5 | 28.16 | 27.58 | 27.20 | 27.59 | 28.95 | 29.10 | *29.25* | *29.25* | 28.87 | **29.47** |
| | LIVE1 | 26.98 | 26.07 | 26.06 | 26.12 | 27.95 | 27.99 | *28.10* | *28.10* | 27.82 | **28.36** |
| | BSD68 | 26.81 | 25.86 | 25.75 | 26.01 | 27.91 | 27.91 | *27.95* | *27.95* | 27.76 | **28.19** |

## 4.2 Experiments on Synthetic Non-I.I.D. Gaussian Noise Cases

Similar to [46], we collected a set of source images to train the network, including 432 images from BSD [5], 400 images from the validation set of ImageNet [12] and 4744 images from the Waterloo Exploration Database [26]. Three commonly used datasets in image restoration (Set5, LIVE1 and BSD68 in [20]) were adopted as test datasets to evaluate the performance of different methods. In order to evaluate the effectiveness and robustness of VDN under the non-i.i.d. noise configuration, we simulated the non-i.i.d. Gaussian noise as following,

$$\boldsymbol{n} = \boldsymbol{n}^1 \odot \boldsymbol{M}, \ \ n_{ij}^1 \sim \mathcal{N}(0,1), \tag{14}$$

where $\boldsymbol{M}$ is a spatially variant map with the same size as the source image. We totally generated four kinds of $\boldsymbol{M}$s as shown in Fig. 2. The first (Fig. 2 (a)) is used for generating noisy images of training data and the others (Fig. 2 (b)-(d)) generating three groups of testing data (denotes as Cases 1-3). Under this noise generation manner, the noises in training data and testing data are with evident difference, suitable to verify the robustness and generalization capability of competing methods.

**Comparson with the State-of-the-art:** Table 1 lists the average PSNR results of all competing methods on three groups of testing data. From Table 1, it can be easily observed that: 1) The VDN outperforms other competing methods in all cases, indicating that VDN is able to handle such complicated non-i.i.d. noise; 2) VDN surpasses FFDNet about 0.25dB averagely even though FFDNet depends on the true noise level information instead of automatically inferring noise distribution as our method; 3) the discriminative methods MLP, DnCNN-B and UDNet seem to evidently overfit on training noise bias; 4) the classical model-driven method CBM3D performs more stably than WNNM and NCSR, possibly due to the latter's improper i.i.d. Gaussian noise assumption. Fig. 3 shows the denoising results of different competing methods on one typical image in testing set of Case 2, and more denoising results can be found in the supplementary material. Note that we only display the top four best results from all due to page limitation. It can be seen that the denoised images by CBM3D and DnCNN-B still contain obvious noise, and FFDNet over-smoothes the image and loses some edge information, while our proposed VDN removes most of the noise and preserves more details.

Even though our VDN is designed based on the non-i.i.d. noise assumption and trained on the non-i.i.d. noise data, it also performs well on additive white Gaussian noise (AWGN) removal task. Table 2 lists the average PSNR results under three noise levels ($\sigma = 15, 25, 50$) of AWGN. It is easy to see that our method obtains the best or at least comparable performance with the state-of-the-art method FFDNet. Combining Table 1 and Table 2, it should be rational to say that our VDN is robust and able to handle a wide range of noise types, due to its better noise modeling manner.

**Noise Variance Prediction:** The *S-Net* plays the role of noise modeling and is able to infer the noise distribution from the noisy image. To verify the fitting capability of *S-Net*, we provided the $\boldsymbol{M}$

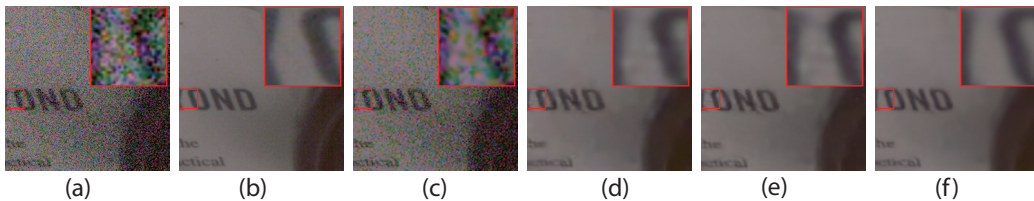

| (a) | (b) | (c) | (d) | (e) | (f) |

Figure 4: Denoising results on one typical image in the validation set of SIDD. (a) Noisy image, (b) Simulated "clean" image, (c) WNNM(21.80dB), (d) DnCNN (34.48dB), (e) CBDNet (34.84dB), (d) VDN (35.50dB).

Table 3: The comparison results of different methods on SIDD benchmark and validation dataset.

| Datasets | SIDD Benchmark | | | | | | SIDD Validation | | |
|---|---|---|---|---|---|---|---|---|---|
| Methods | CBM3D | WNNM | MLP | DnCNN-B | CBDNet | VDN | DnCNN-B | CBDNet | VDN |
| PSNR | 25.65 | 25.78 | 24.71 | 23.66 | 33.28 | **39.23** | 38.41 | 38.68 | **39.28** |
| SSIM | 0.685 | 0.809 | 0.641 | 0.583 | 0.868 | **0.971** | **0.909** | 0.901 | **0.909** |

Table 4: The comparison results of all competing methods on DND benchmark dataset.

| Methods | CBM3D | WNNM | NCSR | MLP | DnCNN-B | FFDNet | CBDNet | VDN |
|---|---|---|---|---|---|---|---|---|
| PSNR | 34.51 | 34.67 | 34.05 | 34.23 | 37.90 | 37.61 | 38.06 | **39.38** |
| SSIM | 0.8507 | 0.8646 | 0.8351 | 0.8331 | 0.9430 | 0.9415 | 0.9421 | **0.9518** |

predicted by *S-Net* as the input of FFDNet, and the denoising results are listed in Table 1 (denoted as FFDNet$_v$). It is obvious that FFDNet under the real noise level and FFDNet$_v$ almost have the same performance, indicating that the *S-Net* effectively captures proper noise information. The predicted noise variance Maps on three groups of testing data are shown in Fig. 2 (b2-d2) for easy observation.

## 4.3 Experiments on Real-World Noise

In this part, we evaluate the performance of VDN on real blind denoising task, including two banchmark datasets: DND [32] and SIDD [1]. DND consists of 50 high-resolution images with realistic noise from 50 scenes taken by 4 consumer cameras. However, it does not provide any other additional noisy and clean image pairs to train the network. SIDD [1] is another real-world denoising benchmark, containing $30,000$ real noisy images captured by 5 cameras under 10 scenes. For each noisy image, it estimates one simulated "clean" image through some statistical methods [1]. About $80\%$ ($\sim 24,000$ pairs) of this dataset are provided for training purpose, and the rest as held for benchmark. And 320 image pairs selected from them are packaged together as a medium version of SIDD, called SIDD Medium Dataset[2], for fast training of a denoiser. We employed this medium vesion dataset to train a real-world image denoiser, and test the performance on the two benchmarks.

Table 3 lists PSNR results of different methods on SIDD benchmark[3]. Note that we only list the results of the competing methods that are available on the official benchmark website[2]. It is evident that VDN outperforms other methods. However, note that neither DnCNN-B nor CBDNet performs well, possibly because they were trained on the other datasets, whose noise type is different from SIDD. For fair comparison, we retrained DnCNN-B and CBDNet based on the SIDD dataset. The performance on the SIDD validation set is also listed in Table 3. Under same training conditions, VDN still outperforms DnCNN-B 0.87 PSNR and CBDNet 0.60dB PSNR, indicating the effectiveness and significance of our non-i.i.d. noise modeling manner. For easy visualization, on one typical denoising example, results of the best four competing methods are displayed in Fig. 4

Table 4 lists the performance of all competing methods on the DND benchmark[4]. From the table, it is easy to be seen that our proposed VDN surpasses all the competing methods. It is worth noting that CBDNet has the same optimized network with us, containing a *S-Net* designed for estimating the noise distribution and a *D-Net* for denoising. The superiority of VDN compared with CBDNet mainly benefits from the deep variational inference optimization.

For easy visualization, on one typical denoising example, results of the best four competing methods are displayed in Fig. 4. Obviously, WNNM is ubable to remove the complex real noise, maybe because the low-rankness prior is insufficient to describe all the image information and the IID Gaussian noise assumption is in conflict with the real noise. With the powerful feature extraction

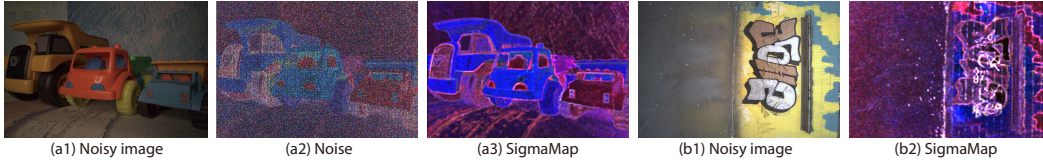

| (a1) Noisy image | (a2) Noise | (a3) SigmaMap | (b1) Noisy image | (b2) SigmaMap |

Figure 5: The noise variance map predicted by our proposed VDN on SIDD and DND benchmarks. (a1-a3): The noisy image, real noise ($|\boldsymbol{y} - \boldsymbol{x}|$) and noise variance map of one typical image of SIDD validation dataset. (b1-b2): The noisy image and predicted noise variance map of one typical image of DND dataset.

Table 5: Performance of VDN under different $\varepsilon_0^2$ values on SIDD validation dataset ($p = 7$).

| $\varepsilon_0^2$ | 1e-4 | 1e-5 | 1e-6 | 1e-7 | 1e-8 | MSE |
|---|---|---|---|---|---|---|
| PSNR | 38.89 | 39.20 | **39.28** | 39.05 | 39.03 | 39.01 |
| SSIM | 0.9046 | 0.9079 | **0.9086** | 0.9064 | 0.9063 | 0.9061 |

Table 6: Performance of VDN under different $p$ values on SIDD validation dataset ($\varepsilon_0^2 = 1e$-6).

| $p$ | 5 | 7 | 11 | 15 | 19 |
|---|---|---|---|---|---|
| PSNR | 39.26 | **39.28** | 39.26 | 39.24 | 39.24 |
| SSIM | **0.9089** | 0.9086 | 0.9086 | 0.9079 | 0.9079 |

ability of CNN, DnCNN and CBDNet obtain much better denoising results than WNNM, but still with a little noise. However, the denoising result of our proposed VDN has almost no noise and is very close to the groundtruth.

In Fig. 5, we displayed the noise variance map predicted by *S-Net* on the two real benchmarks. The variance maps had been enlarged several times for easy visualization. It is easy to see that the predicted noise variance map relates to the image content, which is consistent with the well-known signal-depend property of real noise to some extent.

### 4.4  Hyper-parameters Analysis

The hyper-parameter $\varepsilon_0$ in Eq. (2) determines how much does the desired latent clean image $\boldsymbol{z}$ depend on the simulated groundtruth $\boldsymbol{x}$. As discussed in Section 3.6, the negative variational lower bound degenerates into MSE loss when $\varepsilon_0$ is setted as an extremely small value close to 0. The performance of VDN under different $\varepsilon_0$ values on the SIDD validation dataset is listed in Table 5. For explicit comparison, we also directly trained the *D-Net* under MSE loss as baseline. From Table 5, we can see that: 1) when $\varepsilon_0$ is too large, the proposed VDN obtains relatively worse results since the prior constraint on $\boldsymbol{z}$ by simulated groundtruth $\boldsymbol{x}$ becomes weak; 2) with $\varepsilon_0$ decreasing, the performance of VDN tends to be similar with MSE loss as analysised in theory; 3) the results of VDN surpasses MSE loss about 0.3 dB PSNR when $\varepsilon_0^2 = 1e$-6, which verifies the importantance of noise modeling in our method. Therefore, we suggest that the $\varepsilon_0^2$ is set as $1e$-5 or $1e$-6 in the real-world denoising task.

In Eq. (3), we introduced a conjugate inverse gamma distribution as prior for $\boldsymbol{\sigma}^2$. The mode of this inverse gamma distribution $\xi_i$ provides a rational approximate evaluation for $\sigma_i^2$, which is a local estimation in a $p \times p$ window centered at the $i^{th}$ pixel. We compared the performance of VDN under different $p$ values on the SIDD validation dataset in Table 6. Empirically, VDN performs consistently well for the hyper-parameter $p$.

## 5  Conclusion

We have proposed a new variational inference algorithm, namely varitional denoising network (VDN), for blind image denoising. The main idea is to learn an approximate posterior to the true posterior with the latent variables (including clean image and noise variances) conditioned on the input noisy image. Using this variational posterior expression, both tasks of blind image denoising and noise estimation can be naturally attained in a unique Bayesian framework. The proposed VDN is a generative method, which can easily estimate the noise distribution from the input data. Comprehensive experiments have demonstrated the superiority of VDN to previous works on blind image denoising. Our method can also facilitate the study of other low-level vision tasks, such as super-resolution and deblurring. Specifically, the fidelity term in these tasks can be more faithfully set under the estimated non-i.i.d. noise distribution by VDN, instead of the traditional i.i.d. Gaussian noise assumption.

**Acknowledgements**:This research was supported by National Key R&D Program of China (2018YFB1004300), the China NSFC project under contract 61661166011, 11690011, 61603292, 61721002 and U1811461, and Kong Kong RGC General Research Fund (PolyU 152216/18E).

## Footnotes

[1]We use $j \ (= 1, \cdots, n)$ and $i \ (= 1, \cdots, d)$ to express the indexes of training data and data dimension, respectively, throughout the entire paper.

[2]https://www.eecs.yorku.ca/ kamel/sidd/index.php

[3]We employed the function 'compare_ssim' in scikit-image library to calculate the SSIM value, which is a little difference with the SIDD official results

[4]https://noise.visinf.tu-darmstadt.de/

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
