[Supplementary Material · nips_2019_supp.pdf]

# Supplementary Material of "Variational Denoising Network: Toward Blind Noise Modeling and Removal"

## Abstract

In this supplementary material, we provide more calculation details on the deduction of the variational lower bound, and demonstrate more experimental results in blind image denoising.

## 1 Calculation Details on the Variational Lower Bound

### 1.1 Model Formation

Let's denote $\boldsymbol{y} \in \mathbb{R}^d$ as the observed noisy image and $\boldsymbol{z} \in \mathbb{R}^d$ the latent clean image. Different from most of the traditional methods, we assumed the noise is distributed as non-i.i.d. Gaussian distribution, i.e.,

$$y_i \sim \mathcal{N}(y_i|z_i, \sigma_i^2), \ i = 1, 2, \cdots, d, \tag{1}$$

where $\mathcal{N}(\cdot|\mu, \sigma^2)$ represents the Gaussian distribution with mean $\mu$ variance $\sigma^2$.

The simulated clean image $\boldsymbol{x}$ evidently provides a strong prior to the latent variable $\boldsymbol{z}$. Accordingly we impose the following conjugate Gaussian prior on $\boldsymbol{z}$:

$$z_i \sim \mathcal{N}(z_i|x_i, \varepsilon_0^2), \ i = 1, 2, \cdots, d, \tag{2}$$

where $\varepsilon_0$ is a hyper-parameter and can be easily set as a small value.

Besides, for $\boldsymbol{\sigma}^2 = \{\sigma_1^2, \sigma_2^2, \cdots, \sigma_d^2\}$, we also introduce a rational conjugate prior as follows:

$$\sigma_i^2 \sim \text{IG}\left(\sigma_i^2 \Big| \frac{p^2}{2} - 1, \frac{p^2 \xi_i}{2}\right), \ i = 1, 2, \cdots, d, \tag{3}$$

where $\text{IG}(\cdot|\alpha, \beta)$ is the inverse gamma distribution with parameter $\alpha$ and $\beta$, $\boldsymbol{\xi} = \mathcal{G}\left((\hat{\boldsymbol{y}} - \hat{\boldsymbol{x}})^2; p\right)$ represents the filtering output of the variance map $(\hat{\boldsymbol{y}} - \hat{\boldsymbol{x}})^2$ by a Gaussian filter with $p \times p$ window, $\hat{\boldsymbol{y}}, \hat{\boldsymbol{x}} \in \mathbb{R}^{h \times w}$ are the matrix (image) forms of $\boldsymbol{y}, \boldsymbol{x} \in \mathbb{R}^d$, respectively. Note that the mode of above IG distribution is $\xi_i$, which is a rational approximate evaluation of $\sigma_i^2$ under $p \times p$ window.

Combining Eqs (1)-(3), a full Bayesian model for the problem can be obtained. The goal then turns to construct a variational strategy to infer the posterior of latent variables $\boldsymbol{z}$ and $\boldsymbol{\sigma}^2$ from noisy image $\boldsymbol{y}$, i.e., $p(\boldsymbol{z}, \boldsymbol{\sigma}^2|\boldsymbol{y})$.

### 1.2 Variational Lower Bound

Instead of calculating the posteriori $p(\boldsymbol{z}, \boldsymbol{\sigma}^2|\boldsymbol{y})$ directly, we introduced another distribution $q(\boldsymbol{z}, \boldsymbol{\sigma}^2|\boldsymbol{y})$ to approximate it. Based on such approximate distribution, we can decompose the marginal likelihood

24   of $\boldsymbol{y}$ as follows:

$$
\begin{aligned}
\log p(\boldsymbol{y}; z, \boldsymbol{\sigma}^2) &= \int q(\boldsymbol{z}, \boldsymbol{\sigma}^2|\boldsymbol{y}) \log p(\boldsymbol{y}|z, \boldsymbol{\sigma}^2) \, \mathrm{d}\boldsymbol{z} \, \mathrm{d}\boldsymbol{\sigma}^2 \\
&= \int q(\boldsymbol{z}, \boldsymbol{\sigma}^2|\boldsymbol{y}) \log \left[ \frac{p(\boldsymbol{y}|z, \boldsymbol{\sigma}^2)p(\boldsymbol{z})p(\boldsymbol{\sigma}^2)}{p(\boldsymbol{z}, \boldsymbol{\sigma}^2|\boldsymbol{y})} \right] \mathrm{d}\boldsymbol{z} \, \mathrm{d}\boldsymbol{\sigma}^2 \\
&= \int q(\boldsymbol{z}, \boldsymbol{\sigma}^2|\boldsymbol{y}) \log \left[ \frac{p(\boldsymbol{y}|z, \boldsymbol{\sigma}^2)p(\boldsymbol{z})p(\boldsymbol{\sigma}^2)}{q(\boldsymbol{z}, \boldsymbol{\sigma}^2|\boldsymbol{y})} + \frac{q(\boldsymbol{z}, \boldsymbol{\sigma}^2|\boldsymbol{y})}{p(\boldsymbol{z}, \boldsymbol{\sigma}^2|\boldsymbol{y})} \right] \right] \mathrm{d}\boldsymbol{z} \, \mathrm{d}\boldsymbol{\sigma}^2 \\
&= \int q(\boldsymbol{z}, \boldsymbol{\sigma}^2|\boldsymbol{y}) \log \left[ \frac{p(\boldsymbol{y}|z, \boldsymbol{\sigma}^2)p(\boldsymbol{z})p(\boldsymbol{\sigma}^2)}{q(\boldsymbol{z}, \boldsymbol{\sigma}^2|\boldsymbol{y})} \right] \mathrm{d}\boldsymbol{z} \, \mathrm{d}\boldsymbol{\sigma}^2 \\
&\qquad\qquad + \int q(\boldsymbol{z}, \boldsymbol{\sigma}^2|\boldsymbol{y}) \log \left[ \frac{q(\boldsymbol{z}, \boldsymbol{\sigma}^2|\boldsymbol{y})}{p(\boldsymbol{z}, \boldsymbol{\sigma}^2|\boldsymbol{y})} \right] \mathrm{d}\boldsymbol{z} \, \mathrm{d}\boldsymbol{\sigma}^2 \\
&= E_{q(\boldsymbol{z}, \boldsymbol{\sigma}^2|\boldsymbol{y})} \left[ \log p(\boldsymbol{y}|z, \boldsymbol{\sigma}^2)p(\boldsymbol{z})p(\boldsymbol{\sigma}^2) - \log q(\boldsymbol{z}, \boldsymbol{\sigma}^2|\boldsymbol{y}) \right] \\
&\qquad\qquad + D_{KL}(q(\boldsymbol{z}, \boldsymbol{\sigma}^2|\boldsymbol{y})||p(\boldsymbol{z}, \boldsymbol{\sigma}^2|\boldsymbol{y})). \quad (4)
\end{aligned}
$$

25   The secode term is a KL divergence of the approximation $q(\boldsymbol{z}, \boldsymbol{\sigma}^2|\boldsymbol{y})$ to the true posterior $p(\boldsymbol{z}, \boldsymbol{\sigma}^2|\boldsymbol{y})$,
26   which is non-negative, and thus the first term constitutes a *variational lower bound* on the marginal
27   likelihood of $p(\boldsymbol{y}|z, \boldsymbol{\sigma}^2)$, i.e.,

$$
\begin{aligned}
\log p(\boldsymbol{y}; z, \boldsymbol{\sigma}^2) &\geq \mathcal{L}(\boldsymbol{z}, \boldsymbol{\sigma}^2; \boldsymbol{y}) \\
&= E_{q(\boldsymbol{z}, \boldsymbol{\sigma}^2|\boldsymbol{y})} \left[ \log p(\boldsymbol{y}|z, \boldsymbol{\sigma}^2)p(\boldsymbol{z})p(\boldsymbol{\sigma}^2) - \log q(\boldsymbol{z}, \boldsymbol{\sigma}^2|\boldsymbol{y}) \right]. \quad (5)
\end{aligned}
$$

28   Similar to the traditional mean-field variation methods, we assumed the independence between
29   variable $\boldsymbol{z}$ and $\boldsymbol{\sigma}^2$, i.e.,

$$
q(\boldsymbol{z}, \boldsymbol{\sigma}^2|\boldsymbol{y}) = q(\boldsymbol{z}|\boldsymbol{y})q(\boldsymbol{\sigma}^2|\boldsymbol{y}). \quad (6)
$$

30   Based on the conjugate priors in Eq. 2 and 3, it is natural to formulate variational posterior forms of
31   $\boldsymbol{z}$ and $\boldsymbol{\sigma}^2$ as follows:

$$
q(\boldsymbol{z}|\boldsymbol{y}) = \prod_i^d \mathcal{N}(z_i|\mu_i(\boldsymbol{y}; W_D), m_i^2(\boldsymbol{y}; W_D)), \; q(\boldsymbol{\sigma}^2|\boldsymbol{y}) = \prod_i^d \mathrm{IG}(\sigma_i^2|\alpha_i(\boldsymbol{y}; W_S), \beta_i(\boldsymbol{y}; W_S)), \quad (7)
$$

32   where $\mu_i(\boldsymbol{y}; W_D)$ and $m_i^2(\boldsymbol{y}; W_D))$ are designed as the prediction functions for getting posterior
33   parameters of latent variable $\boldsymbol{z}$ directly from $\boldsymbol{y}$. The function is represented as a network, called
34   denoising network or *D-Net*, with parameters $W_D$. Similarly, $\alpha_i(\boldsymbol{y}; W_S)$ and $\beta_i(\boldsymbol{y}; W_S))$ denote
35   the prediction functions for evaluating posterior parameters of $\boldsymbol{\sigma}^2$ from $\boldsymbol{y}$, where $W_S$ represents the
36   parameters of a network, called Sigma network or *S-Net*, for predicting them. Our aim is then to
37   optimize these two network parameters $W_D$ and $W_S$ so as to get the explicit functions for predicting
38   clean image variable $\boldsymbol{z}$ as well as noise knowledge $\boldsymbol{\sigma}^2$ from any test noisy image $\boldsymbol{y}$. A rational
39   objective function with respect to $W_D$ and $W_S$ is thus necessary for using gradient decent strategies
40   to train both networks.

41   For notation convenience, we simply write $\mu_i(\boldsymbol{y}; W_D), m_i^2(\boldsymbol{y}; W_D)), \alpha_i(\boldsymbol{y}; W_S), \beta_i(\boldsymbol{y}; W_S))$ as $\mu_i$,
42   $m_i^2$, $\alpha_i$, $\beta_i$ in the following calculations.

43   Combining Eqs (5), (6) and Eq (7), the lower bound can be rewritten as:

$$
\mathcal{L}(\boldsymbol{z}, \boldsymbol{\sigma}^2; \boldsymbol{y}) = E_{q(\boldsymbol{z}, \boldsymbol{\sigma}^2|\boldsymbol{y})} \left[ \log p(\boldsymbol{y}|z, \boldsymbol{\sigma}^2) \right] - D_{KL} \left( q(\boldsymbol{z}|\boldsymbol{y})||p(\boldsymbol{z}) \right) - D_{KL} \left( q(\boldsymbol{\sigma}^2|\boldsymbol{y})||p(\boldsymbol{\sigma}^2) \right), \quad (8)
$$

44 Next we calculated the three terms in Eq (8) one by one as follows:

$$
\begin{aligned}
E_{q(\boldsymbol{z},\boldsymbol{\sigma}^2|\boldsymbol{y})}\left[\log p(\boldsymbol{y}|\boldsymbol{z},\boldsymbol{\sigma}^2)\right] &= \int q(\boldsymbol{z},\boldsymbol{\sigma}^2|\boldsymbol{y})\log p(\boldsymbol{y}|\boldsymbol{z},\boldsymbol{\sigma}^2)\,\mathrm{d}\boldsymbol{z}\,\mathrm{d}\boldsymbol{\sigma}^2 \\
&= \sum_i^n \int q(z_i,\sigma_i^2|\boldsymbol{y})\log p(y_i|z_i,\sigma_i^2)\,\mathrm{d}z_i\,\mathrm{d}\sigma_i^2 \\
&= \sum_i^n \int q(z_i|\boldsymbol{y})q(\sigma_i^2|\boldsymbol{y})\left\{-\frac{1}{2}\log 2\pi - \frac{1}{2}\log\sigma_i^2 - \frac{(y_i-z_i)^2}{2\sigma_i^2}\right\}\mathrm{d}z_i\,\mathrm{d}\sigma_i^2 \\
&= \sum_i\left\{-\frac{1}{2}\log 2\pi - \frac{1}{2}\int q(\sigma_i^2|\boldsymbol{y})\log\sigma_i^2\,\mathrm{d}\sigma_i^2 \int q(z_i|\boldsymbol{y})\,\mathrm{d}z_i \right.\\
&\qquad\qquad \left. -\frac{1}{2}\int q(z_i|\boldsymbol{y})(y_i-z_i)^2\,\mathrm{d}z_i \int q(\sigma_i^2|\boldsymbol{y})\frac{1}{\sigma_i^2}\,\mathrm{d}\sigma_i^2\right\} \\
&= \sum_i^n\left\{-\frac{1}{2}\log 2\pi - \frac{1}{2}E\left[\log\sigma_i^2\right] - \frac{1}{2}E\left[(y_i-z_i)^2\right]E\left[\frac{1}{\sigma_i^2}\right]\right\} \\
&= \sum_i^n\left\{-\frac{1}{2}\log 2\pi - \frac{1}{2}(\log\beta_i - \psi(\alpha_i)) - \frac{\alpha_i}{2\beta_i}\left[(y_i-\mu_i)^2 + m_i^2\right]\right\},
\end{aligned}
\tag{9}
$$

$$
\begin{aligned}
D_{KL}(q(\boldsymbol{z}|\boldsymbol{y})||p(\boldsymbol{z})) &= \sum_i^n D_{KL}(\mathcal{N}(z_i|\mu_i,m_i^2)||p(z_i|x_i,\varepsilon_0^2)) \\
&= \sum_i^n\left\{\frac{(\mu_i-x_i)^2}{2\varepsilon_0^2} + \frac{1}{2}\left[\frac{m_i^2}{\varepsilon_0^2} - \log\frac{m_i^2}{\varepsilon_0^2} - 1\right]\right\},
\end{aligned}
\tag{10}
$$

$$
\begin{aligned}
D_{KL}\left(q(\boldsymbol{\sigma}^2|\boldsymbol{y})||p(\boldsymbol{\sigma}^2)\right) &= \sum_i^n D_{KL}\left(\mathrm{IG}(\sigma_i^2|\alpha_i,\beta_i)||\mathrm{IG}\left(\sigma_i^2|\frac{p^2}{2}-1,\frac{p^2\xi_i}{2}\right)\right) \\
&= \sum_i^n\left\{\left(\alpha_i - \frac{p^2}{2} + 1\right)\psi(\alpha_i) + \left[\log\Gamma\left(\frac{p^2}{2}-1\right) - \log\Gamma(\alpha_i)\right] \right.\\
&\qquad \left. + \left(\frac{p^2}{2}-1\right)\left(\log\beta_i - \log\frac{p^2\xi_i}{2}\right) + \alpha_i\left(\frac{p^2\xi_i}{2\beta_i}-1\right)\right\},
\end{aligned}
\tag{11}
$$

47 Where $\psi(\cdot)$ denotes the digamma function, $E[\cdot]$ represents exception with some stoachastic variables
48 that had been neglected for notation clearity.

49 We can then easily get the expected objective function (i.e., a negtive lower bound of the marginal
50 likelihood on entire training set) for optimizing the network parameters of D-Net and S-Net as follows:
51

$$
\min_{W_D,W_S} -\sum_{j=1}^n \mathcal{L}(\boldsymbol{z}_j,\boldsymbol{\sigma}_j^2;\boldsymbol{y}_j).
\tag{12}
$$

## 52 2 More Experimental Results

### 53 2.1 Experiments on Synthetic Non-I.I.D. Gaussin Noise

54 In this supplementary material, we dispalyed more denoising results of different methods on the
55 testing dataset in Fig. 1-6.

### 56 2.2 Experiments on Real-World Noise

57 In Fig. 7, we show more denoising results of differen methods on the SIDD validation dataset.

Figure 1: Image denoising results of different methods on the testing data in Case 1. From left to right: (a) Noisy Image, (b) Groundtruth, (c) CBM3D, (d) DnCNN-B, (e) FFDNet, (f) VDN

Figure 2: Image denoising results of different methods on the testing data in Case 1. From left to right: (a) Noisy Image, (b) Groundtruth, (c) CBM3D, (d) DnCNN-B, (e) FFDNet, (f) VDN

Figure 3: Image denoising results of different methods on the testing data in Case 2. From left to right: (a) Noisy Image, (b) Groundtruth, (c) CBM3D, (d) DnCNN-B, (e) FFDNet, (f) VDN

Figure 4: Image denoising results of different methods on the testing data in Case 2. From left to right: (a) Noisy Image, (b) Groundtruth, (c) CBM3D, (d) DnCNN-B, (e) FFDNet, (f) VDN

Figure 5: Image denoising results of different methods on the testing data in Case 3. From left to right: (a) Noisy Image, (b) Groundtruth, (c) CBM3D, (d) DnCNN-B, (e) FFDNet, (f) VDN

Figure 6: Image denoising results of different methods on the testing data in Case 3. From left to right: (a) Noisy Image, (b) Groundtruth, (c) CBM3D, (d) DnCNN-B, (e) FFDNet, (f) VDN

Figure 7: Image denoising results of different methods on the SIDD validation set. From left to right: (a) Noisy image, (b) Simulated "clean" image, (c) WNNM, (d) DnCNN-B, (e) CBDNet, (f) VDN