[Reviews · NeurIPS 2019]

Reviewer 1



The idea of parameterizing Bayesian framework with neural networks is very interesting. The learned framework has the benefit of efficient inference as well as good generalization. Paper is overall clearly presented but more implementation details are needed in order for others to reproduce. Detailed comments in the following: 1, At the end, the paper learns a feedforward image denoising neural network. It highly depends on the training dataset. So it is not truly a blind image denoising algorithm, but reviewer won't be too critical about it. In this sense, how does the algorithm perform compared with other recent deep learning denoising algorithms (e.g., MemNet, NLRN) straightly trained on the datasets? 2. It drops the assumption of i.i.d. Guassian distribution for noise. In practice, the noise is closely related to pixel illuminance, but this is not modeled in the framework. 3. In essence, because it learned a feedforward neural network, it looks to the reviewer that, what the algorithm really does is to restrict the field of view of the network in order to have a good estimation of the local noise. Would like to hear comments from the authors about it. 4. In the experiment section, the paper does not report ablation study for hyper-parameter epsilon_0. Specifically, setting epsilon_0 to be zero (simple MSE loss) should be the baseline for the proposed algorithm. How sensitive the algorithm is to epsilon_0? 5. Any noise variance visualization for the real dataset? Can one spot some patterns?

Reviewer 2



Originality: This work presents a novel variational approach for image denoising. Quality: The submission describes a variational approach for image denoising task in the classical blind denoising setting, where both clean and noisy data are available for training. They especially consider the case where the noise is non- i.i.d. Gaussian. The authors have quite careful evaluation of their method compared to related work. For instance, they re-train the compared methods from related work in the same setting as theirs in order to make a relevant comparison. Clarity: Could the authors further discuss the choice of the inverse Gamma prior for the noise variance. Additionally G function, that utilizes the p x p window should be discussed further. The authors should discuss the choices of p along with empirical evaluation. Significance: The method is novel and the empirical results are good. Improvements over related work are consistent but not huge e.g., in the real noisy data setting. The best results are in the synthetic setting which exactly matches with the setup of the model (spatially varying non-i.i.d. Gaussian noise), but it is not discussed whether this setting is the best for real-world noisy data. It is likely that researchers will take this approach and fine-tune it to other scenarios as well in future work. The authors mention that the network architectures for S- and D-Nets can be replaced with others, but it would have been important to see what is the effect of the chosen architectures. Since the input and output dimensions are similar for both nets, it would be possible to test all combinations (both with U-Net, both with DnCNN architecture and other mixtures). This would allow to reader to assess the contributions of the network architectures vs. the loss functions used in training. It would have been interesting to see empirical evaluation of different noise types, for instance signal dependent noise, correlated noise, at least in the synthetic noise setting. Minor issues: There are spelling errors in the document that should be fixed. For instance: - In section 1: The word "implementation" is ambiguous in this setting. Consider something along these lines: “relatively low implementation speed” -> “low runtime speed”, “Needs to be re-implemented” -> “Needs to be re-optimized”. - 3.3 "It’s pleased that all the three terms" -> "Fortunately all the three terms". - 3.3. line 154 and supplementary line 50 "negtive lower" -> negative lower. - 3.6 title “Some Discussions” -> “Discussion”. - 3.6 line 189 “our method is a generative approach” -> “our method is a generative one”. A minor issue with clarity resides with Tables 1 and 2, namely the second best entry marked with italics is hard to differentiate from normal font, please consider using e.g., parenthesis instead.

Reviewer 3



Originality: While the variational formulation for the given image denoising is new, it is not clear why some of the choices for the prior distribution is valid. (e.g., eq.(4)) While the new formulation makes sense, there are several critical questions regarding clarity and quality of the presentation of the paper. I would like to see the answers for these questions in the rebuttal. - The existence of the simulated clean x is most puzzling. It is not clear how we can always obtain those. Also, if those simulated clean x is available, what is the performance of the direct supervised model that maps y to x? Do we really need the complex variational formulation of the paper? Also, for real-world data, how can one obtain x? I think obtaining x in [1] is a special case since there are multiple noisy observations for the same underlying clean image. But, in more realistic case, it is nearly impossible - Figure 2 is very puzzling. As far as I understand, both D-net and S-net are fixed once the training is done. How can S-net predict the totally new noise pattern that has not been seen during the training? Unless some additional process is described, e.g., fine-tuning with the test data, Figure 2 is not clear to me. - The PSNR numbers for BSD68 (e.g., for sigma=25) in Table 2 is very high. Are all PSNR numbers reproduced by the authors? ==== The rebuttals mostly clarified my questions. Still Fig.2 is a bit surprising result to me, but the generative nature of the Bayesian framework perhaps enables it. I increased my score to 6.

[Author Response · NeurIPS 2019]

**Table 1:** Average results on the three cases of NIID Gaussian Noise (BSD68).

| Methods | DnCNN | FFDNet | MemNet | NLRN | VDN |
|---|---|---|---|---|---|
| PSNR | 28.43 | 28.55 | 28.52 | 28.61 | **28.72** |

**Table 2:** PSNR results under different $\varepsilon_0^2$ values on Renoir Dataset ($p$=11).

| $\varepsilon_0^2$ | 1e-4 | 1e-5 | 1e-6 | 1e-7 | 1e-8 | 0 |
|---|---|---|---|---|---|---|
| PSNR | 38.81 | 39.39 | **39.45** | 39.42 | 39.27 | 39.18 |

**Table 3:** PSNR results of different $p$ values on Renoir Dataset ($\varepsilon_0^2$ =1e-6).

| $p$ | 7 | 11 | 15 | 19 | 23 |
|---|---|---|---|---|---|
| PSNR | 39.36 | **39.45** | 39.39 | 39.20 | 39.06 |

**Table 4:** PSNR results of different architecture combinations on Renoir Dataset.

| Combinations | D-0 | D-U | D-D | U-0 | U-D | U-U |
|---|---|---|---|---|---|---|
| PSNR | 38.51 | **38.80** | 38.67 | 39.18 | **39.45** | 39.35 |

**Table 5:** PSNR and SSIM results on DND and SIDD real Benchmarks.

| Datasets | Metrics | Methods | | | |
|---|---|---|---|---|---|
| | | DnCNN | FFDNet | CBDNet | VDN |
| DND | PSNR | 37.90 | 37.61 | 38.06 | **38.35** |
| | SSIM | 0.9430 | 0.9415 | 0.9421 | **0.9514** |
| SIDD | PSNR | 38.65 | - | 38.68 | **39.04** |
| | SSIM | 0.9089 | - | 0.9093 | **0.9151** |

(a) Noisy image     (c) Real noise     (d) Variance map

**Figure 1:** Estimated variance map of one typical real image in SIDD Dataset.

**Table 6:** PSNR results of different methods under the IID Gaussian noise on the gray BSD68 Dataset.

| Metric | $\sigma = 15$ | | | | | $\sigma = 25$ | | | | | $\sigma = 50$ | | | | |
|---|---|---|---|---|---|---|---|---|---|---|---|---|---|---|---|
| | BM3D | DnCNN | FFDNet | UDNet | VDN | BM3D | DnCNN | FFDNet | UDNet | VDN | BM3D | DnCNN | FFDNet | UDNet | VDN |
| PSNR | 31.06 | 31.60 | **31.63** | 31.39 | 31.61 | 28.56 | 29.18 | 29.16 | 28.84 | **29.23** | 25.66 | 26.28 | 26.35 | 26.02 | **26.43** |

**To Reviewer 1:**

**Q1.1 Compared with other DL algorithms:** As suggested, we have re-trained additional STOA DL methods MemNet and NLRN on our datasets, and listed the PSNR results in Tabel 1. We'll additionally cite the related references and add results in revision.

**Q1.2&1.5 Pixel illuminance related noise, noise variance visualization for real dataset:** Actually, this is one reason why we drop conventional i.i.d. noise assumption. Even not explicitly modeling such signal dependent property, our method can finely fit spatially variant noise, as explained in **Q1.3**, more or less delivering this noise property. As shown in Fig. 1 (noise variance by our method), our method is capable of estimating signal dependent noises, complying with practical understanding for real image noises.

**Q1.3 Good estimation of local noise:** Thanks for understanding and illuminating this point. On one hand, the mode of the inverse Gamma distribution in Eq. (4) is locally estimated in different image space. On the other hand, *S-Net* predicts the noise variance for each pixel via the those located in its local receptive field. Compared with conventional filtering-based variance estimation methods with pre-designed filters, our method employs a learnable CNN to adaptively fit filters for different local areas. This naturally conducts its capability on spatially variant noise estimation and robust denoising effect.

**Q1.4 Sensitivity to $\varepsilon_0$:** We have tested the sensitivity of $\varepsilon_0^2$ (Table 2 shows results on Renoir data), showing that our method performs stably well when setting it in around [1e-7,1e-5]. $\varepsilon_0^2 = 0$ denotes the network directly trained under the MSE loss as conventional.

**To Reviewer 2:**

**Q2.1 Choice of inverse Gamma (IG) prior.** IG is adopted because it is the conjugate prior for the variance of Gaussian distribution, enabling the posterior of $\sigma^2$ analytically calculated. The function of the Gaussian filter with $p \times p$ window is to make IG parameters capable of being estimated locally in different image space, so as to enable the method deliver spatial noise variations, as discussed in **Q1.3**. Empirically, our method can perform consistently well for $p \in [7, 15]$. A typical example is given in Table 3.

**Q2.2 Effect of network architectures.** As suggested, we have tested different network architecture combinations on Renoir Dataset (Table 4), including those obtained by both with U-Net (U-U) or DnCNN (D-D), mixture of the two (U-D and D-U), and only training one under MSE loss (U-0 and D-0). It is seen that our method is not too sensitive to the chosen network architectures, and using the designed loss function can evidently improve the denoising effect beyond directly training the network as conventional.

**Q2.3 Signal dependent noise.** As discussed in **Q1.2**, the real-world noisy images (e.g., Fig. 1) used in the paper actually contain signal dependent noise, and the better performance of our method on them verifies its effectiveness on such typical real noises.

**Q2.4 SSIM.** We list SSIM comparison in Table 5, corresponding to Table 4&5 of the paper. Superiority of our method is also evident.

**Q2.5 Usable in other scenarios.** Yes. Just as other known denoising methods, our method can also be easily embedded into other low-level tasks, like super-resolution and deblurring. We will extend our model to other scenarios in our future investigations.

**To Reviewer 3:**

**Q3.1 How can obtain x.** Please kindly note that our problem setting is exactly the same as current supervised deep learning (DL) methods, representing the present STOA methodology for the image denoising task. All of them are trained on a pre-collected supervised dataset composing of noisy-clean image pairs $\{y_i, x_i\}$s, which are either simulated according to the in-camera pipeline, or elaborately collected in the recent real-world image denoising datasets, like Renoir and SIDD we have employed. We thus just follow what the other DL methods did. In the paper, we have actually compared with several direct supervised DL methods and two additional ones MemNet and NLRN (**Q1.1**). The advantage of our method is clear, especially in more practical non-i.i.d. noise cases. The superiority of our method actually just lies in the designing of the loss function, i.e., a variational lower bound for posterior inference (just like that of VAE compared with conventional autoencoder). It makes our model a generative one constructed under Bayesian framework with better interpretability and generality, as validated by our experiments and also evidently shown in **Q2.2**. We thus strongly believe that such a variational formulation (the main contribution of this work) should be necessary and important. We sincerely ask the reviewer to further read our descriptions on the motivation and insights of this work (Sec. 3) and the more explanations on our model provided in **Q1.3** and **Q2.2**, and reconsider the rating on this work. Thanks.

**Q3.2 How S-net predicts new noise.** Our method does not need additional fine-tune or other processing in the test stage. All noise predictions, including those never seen in training (as shown in Fig.2), are directly got using the fixed *S-Net* obtained in the training stage. Such generalization capability is naturally attributed to the non i.i.d. noise modeling mechanism and generative insights of our model under the Bayesian framework, in which the noise of each pixel is locally estimated by the learned *S-Net* as discussed in **Q1.3**.

**Q3.3 High PSNR for BSD68.** Results listed in Table 2 are obtained on sRGB images while not gray ones. Due to mutual correction among RGB channels, the results tend to be higher than those obtained on gray images. To make this clearer, we also list typical performance of our method on gray images of BSD68 Dataset in Table 6, more complying with the results reported in other works.

[Meta-Review · NeurIPS 2019]

The paper initially received two positive and one negative review. Later, the negative reviewer was convinced by the rebuttal and increased his score. Overall, all reviewers reached a consensus that the paper should be accepted. The area chair agrees with their assessment and follows their recommendation.